# DAWI: DUAL ANCHORED WEIGHTED INTERPOLATION FOR LLM UNLEARNING

## ABSTRACT

Large Language Models (LLMs) are trained on vast amounts of text data, and they frequently memorize sensitive or private information that appears in the training corpus. This raises significant privacy, security, and ethical concerns, particularly when such information can be extracted by adversarial prompts or from membership inference attacks. Machine unlearning has therefore emerged as an important research direction, with the goal of selectively removing knowledge of problematic information from a model while preserving its general language understanding and reasoning capabilities. In this work, we focus on unlearning information that is memorized during the fine-tuning phase of training, which commonly happens when inference providers fine-tune models on user interactions. We introduce Dual Anchored Weighted Interpolation (DAWI), a simple yet effective unlearning algorithm that achieves state-of-the-art results on the TOFU benchmark and demonstrates strong unlearning efficacy with minimal hyperparameter tuning, making it practical for real-world deployment.

## 1 INTRODUCTION

Large language models (LLMs) often memorize training data, giving rise to concerns about privacy and security (Carlini et al., 2023). However, retraining from scratch is impractical due to the computational cost of LLM training. Machine unlearning (Cao & Yang, 2015) aims to address this problem by removing the influence of specific training examples without requiring a full retraining of the model. Conceptually, the most reliable way to ensure forgetting is exact unlearning, in which the model is retrained from scratch on the dataset with the sensitive entries removed. However, the scale of modern LLMs makes this strategy computationally prohibitive, leading to the development of approximate unlearning methods, with the goal of removing the influence of undesirable information in a computationally efficient manner.

In this paper, we specifically study the case of unlearning where an LLM memorizes information after pretraining. This reflects common deployment pipelines, where inference providers fine-tune models on user interactions with methods such as RLHF (Ouyang et al., 2022) to improve overall performance. Users may accidentally send sensitive information that they would like to be removed, such as passwords or classified information, to the model. This issue is intensified by privacy laws such as the European Union's General Data Protection Regulation (GDPR), which require organizations to remove users' information upon request.

Current unlearning methods are designed for general unlearning tasks and typically assume access to a fine-tuned model (Mekala et al., 2025; Fan et al., 2024; Chen & Yang, 2023), without access to a prior iteration of the model that is free from the influence of the undesirable data, which we refer to as the base model. However, in the real world, model providers regularly save model checkpoints, so our work explores additional improvements that can be made with the addition of a base model.

We propose Dual Anchored Weighted Interpolation (DAWI), a flexible unlearning method that does not use a traditional optimizer and achieves strong results on a variety of unlearning tasks. DAWI constrains unlearning to per-parameter line segments between a base model and its fine-tuned variant. By reparameterizing each weight as a convex combination of the base and fine-tuned models' weights, and sparsely updating only the mixing coefficients, DAWI performs quantized optimization of the unlearning objective while staying inside the convex hull of the two models. This directional

anchoring reduces drift, limits over-unlearning, and yields automatically annealed updates through discrete step sizes.

Most unlearning benchmarks to date, such as TOFU (Maini et al., 2024) and MUSE (Shi et al., 2024), measure a model's utility after unlearning by testing its ability to answer short, single-turn questions, either from general knowledge or a small retain set. While useful for initial comparisons, these evaluations fail to capture the complexity of real-world deployments, where models are often required to perform multi-step reasoning and sequential decision-making to accomplish tasks.

To bridge this gap, we introduce Math Unlearning Dataset (MUD), which consists of two splits, MUD-200 and MUD-20k, with 200 and 20k data points, respectively. Unlike existing unlearning datasets, MUD is explicitly designed to test whether a model can still retain complex reasoning ability after unlearning. After training, we evaluate model utility on GSM8k (Cobbe et al., 2021), a benchmark requiring chain-of-thought style reasoning in mathematical problem solving. This setup better reflects the demands of practical applications, where preserving higher-order reasoning capabilities is just as important as ensuring effective forgetting.

Our primary contribution is DAWI, a practical and novel unlearning approach, which reaches state-of-the-art performance on the TOFU benchmark with minimal hyperparameter tuning. We conduct a variety of experiments to show that DAWI performs well on unlearning tasks and ablation studies to justify design choices.

## 2 RELATED WORK

A variety of machine unlearning approaches have been attempted (Wang et al., 2024), and much recent research explores unlearning in LLMs (Jang et al., 2022; Ji et al., 2024; Lu et al., 2022; Lynch et al., 2024; Patil et al., 2024; Pawelczyk et al., 2023). Unlearning methods are diverse, including model optimization methods, prompt filtering, and inference-time modifications (Thaker et al., 2024; Eldan & Russinovich, 2023; Jia et al., 2024). However, prompt filtering and inference-time modifications were often found to be unreliable and vulnerable to paraphrasing on benchmarks like TOFU.

Initially, unlearning approaches were primarily inspired by gradient ascent, which aims to maximize traditional softmax loss on the forget set (Chen & Yang, 2023; Jang et al., 2022). However, gradient-ascent style "negative fine-tuning" often induces instability and degrades performance on retained knowledge, especially when the forget set is large or semantically diverse, leading to pronounced utility loss (Zhang et al., 2024).

More recently, methods such as Representation Misdirection for Unlearning (RMU) and Negative Preference Optimization (NPO) have achieved substantially higher performance on general unlearning tasks by moving beyond pure loss maximization and shaping model behavior through representation control and preference-based training, respectively (Zhang et al., 2024; Li et al., 2024). RMU seeks to redirect internal representations away from targeted knowledge regions while preserving global competence, whereas NPO takes inspiration from Direct Preference Optimization (Rafailov et al., 2023) and optimizes a smoother loss function. These methods report stronger empirical unlearning alongside improved retention relative to gradient ascent.

However, without a base model, these unlearning methods frequently suffer from over-unlearning, where updates intended to suppress specific information may also impair semantically adjacent or even unrelated capabilities, producing utility drops on the retain set. Moreover, over-unlearning decreases the probability of producing sequences from the forget set well below the baseline, allowing an attacker with access to a model's weights or logits to infer the contents of the forget set (Zhang et al., 2025; Carlini et al., 2023; Shi et al., 2023; Yeom et al., 2018).

Moreover, almost all LLM unlearning methods use standard optimizers, which indiscriminately apply updates to all of a model's parameters. It is unlikely that all of a model's parameters have knowledge about a given datapoint in the forget set; thus, modifying all of the model's weights on every optimizer step may unnecessarily degrade the model's general performance.

Many unlearning methods introduce additional hyperparameters that require additional tuning. NPO and its variants, such as Alternate Preference Optimization (AltPO) and Simple Negative Preference Optimization (SimNPO) (Fan et al., 2024; Mekala et al., 2025), use an $\alpha$ term that controls the

strength of retain loss compared to forget loss, along with a smoothing term, $\beta$. Hyperparameter tuning has a significant impact on the result of unlearning, and each additional hyperparameter exponentially increases the search space; thus, it is desirable to have as few hyperparameters as possible. DAWI introduces one tunable hyperparameter that replaces the learning rate in traditional optimizers. This means that DAWI introduces zero net parameters compared to traditional gradient descent.

---

**Algorithm 1** DAWI, our proposed unlearning algorithm. See Section 3 for a more detailed but less concise explanation. $A_\theta$ denotes the collection of all weight matrices, and $A_\theta^k$ denotes the $k$th weight matrix of $\pi_\theta$.

---

**Require:** $\pi_{\text{fine-tune}}$: Model trained on retain and forget set
**Require:** $\pi_{\text{base}}$: Base model that has not seen the retain or forget set
**Require:** $\pi_\theta$: Model that undergoes training
**Require:** $B^k, C^k$: Matrix accumulators for each weight matrix $A_\theta^k \in A_\theta$
**Require:** $c \in \mathbb{N}$: Hyperparameter that determines step size
**Require:** $E$: Number of training epochs
1: Initialize $B^k \leftarrow \mathbf{0}$, $C^k \leftarrow c\,\mathbf{1}$ for all $k$
2: Initialize $\pi_\theta \leftarrow \pi_{\text{fine-tune}}$
3: **for** $e = 1$ to $E$ **do**
4:     Compute full gradients $G \leftarrow \nabla\mathcal{L}(A_\theta)$
5:     **for** $A_\theta^k \in A_\theta$ **do**         ▷ **Proposal pass:** Estimate $|\Delta\mathcal{L}_{i,j}^k|$ for all entries
6:         $\Delta A^k \leftarrow A_{\text{fine-tune}}^k - A_{\text{base}}^k$
7:         **for all** indices $(i,j)$ in $A_\theta^k$ **do**
8:             $g \leftarrow \frac{\partial\mathcal{L}}{\partial(A_\theta^k)_{i,j}}$
9:             $s \leftarrow g \cdot (\Delta A_{i,j}^k)$
10:            **if** $s \geq 0$ **then**
11:               $\Delta\beta_{i,j}^k \leftarrow \frac{C_{i,j}^k}{C_{i,j}^k + B_{i,j}^k + 1} - \frac{C_{i,j}^k}{C_{i,j}^k + B_{i,j}^k}$
12:            **else**
13:               $\Delta\beta_{i,j}^k \leftarrow \frac{C_{i,j}^k + 1}{C_{i,j}^k + B_{i,j}^k + 1} - \frac{C_{i,j}^k}{C_{i,j}^k + B_{i,j}^k}$
14:            **Estimate** $\Delta\mathcal{L}_{i,j} \leftarrow g \cdot \Delta\beta_{i,j}^k \cdot \Delta A_{i,j}^k$
15:            Append $|\Delta\mathcal{L}_{i,j}|$ to $\Delta\mathcal{L}_{\text{all}}$
16:     $T \leftarrow \frac{1}{5} \cdot Q_{0.99}(\{|\Delta\mathcal{L}_{\text{all}}|\})$
17:     **for** $A_\theta^k \in A_\theta$ **do**         ▷ **Edit pass:** Edit accumulators if $|\Delta\mathcal{L}_{i,j}^k| \geq T$
18:         $\Delta A^k \leftarrow A_{\text{fine-tune}}^k - A_{\text{base}}^k$
19:         **for all** indices $(i,j)$ in $A_\theta^k$ **do**
20:             $g \leftarrow \frac{\partial\mathcal{L}}{\partial(A_\theta^k)_{i,j}}$
21:             $s \leftarrow g \cdot (\Delta A_{i,j}^k)$
22:            **if** $|\Delta\mathcal{L}_{i,j}| \geq T$ **then**
23:               **if** $\Delta\beta_{i,j}^k \geq 0$ **then**
24:                  $C_{i,j}^k \leftarrow C_{i,j}^k + 1$
25:               **else**
26:                  $B_{i,j}^k \leftarrow B_{i,j}^k + 1$
27:            $\alpha_{i,j}^k = \frac{B_{i,j}^k}{C_{i,j}^k + B_{i,j}^k}$
28:            $(A_\theta^k)_{i,j} \leftarrow (1 - \alpha_{i,j}^k)(A_{\text{fine-tune}}^k)_{i,j} + \alpha_{i,j}^k(A_{\text{base}}^k)_{i,j}$

---

## 3 ALGORITHM

### 3.1 PRELIMINARIES AND NOTATION

As is typical for unlearning, we have a set of data points to be removed, known as the forget set ($\mathcal{D}_{\text{forget}}$). In addition, we have a subset of data points that should not be forgotten, known as the retain set ($\mathcal{D}_{\text{retain}}$).

Since we work specifically in the regime where a model learns undesirable information after pre-training, we have access to the base model, $\pi_{\text{base}}$, which has not been trained on the undesirable information, and $\pi_{\text{fine-tune}}$, which is the model that has been trained on a mixture of clean and undesirable data. Our goal is to remove the influence of the undesirable data from the model while retaining the model's performance on the rest of the data.

The traditional goal of machine unlearning is to have the unlearned model, which we denote as $\pi_\theta$, match the performance of the retrained model, $\pi_{\text{retrain}}$, as closely as possible. However, we take a slightly different approach with DAWI. DAWI attempts to match the performance of the fine-tuned model on the retain set and the performance of the base model on the forget set. Since the base model and fine-tuned model are both accessible during training, this can be computed directly.

We denote the base model as $\pi_{\text{base}}$, the fine-tuned model as $\pi_{\text{fine-tune}}$, and the model that is being trained as $\pi_\theta$. We denote the set of all weight matrices of the model $\pi_\theta$ as $A_\theta$, the $k$th weight matrix of $\pi_\theta$ as $(A_\theta^k)$, and an element of that matrix as $(A_\theta^k)_{i,j}$, with similar notation for weight matrices from $\pi_{\text{base}}$ and $\pi_{\text{fine-tune}}$.

We define $\pi_\theta(y \mid x)$ to be the probability of $\pi_\theta$ generating the correct answer, $y$, given the question, $x$, and $\pi_\theta(y_t \mid x, y_{<t})$ to be the probability of $\pi_\theta$ generating the $t$th token of $y$ given its preceding tokens.

### 3.2 INITIALIZATION

For each matrix $A_\theta^k$ in $\pi_\theta$, we initialize two accumulators, $B^k$ and $C^k$, both in the same shape as $A_\theta^k$. We initialize $B_{i,j}^k = 0$, $C_{i,j}^k = c$, where $c$ is a nonzero integer hyperparameter. We define $\alpha_{i,j}^k = \frac{B_{i,j}^k}{C_{i,j}^k + B_{i,j}^k}$, and determine each $(A_\theta^k)_{i,j}$ by linearly interpolating between the base and fine-tuned model.

$$(A_\theta^k)_{i,j} = (1 - \alpha_{i,j}^k)(A_{\text{fine-tune}}^k)_{i,j} + \alpha_{i,j}^k(A_{\text{base}}^k)_{i,j}$$

Thus, during initialization, $\alpha = \mathbf{0}$ and $\pi_\theta = \pi_{\text{fine-tune}}$. For convenience, we define $\beta_{i,j}^k = 1 - \alpha_{i,j}^k$.

### 3.3 LOSS FUNCTION

We use the following simple loss function inspired by reinforcement learning. We define forget loss and retain loss, respectively, as the probability $\pi_\theta(y \mid x)$, clamped token-wise, and raised to the power of $\frac{1}{|y|}$:

$$\mathcal{L}_{\text{forget}}(\theta) = \sum_{(x,y) \in \mathcal{D}_{\text{forget}}} \prod_{t=1}^{|y|} (\frac{\max(\pi_\theta(y_t \mid x, y_{<t}), \pi_{\text{base}}(y_t \mid x, y_{<t}))}{\max(\text{sg}[\pi_\theta(y_t \mid x, y_{<t})], \pi_{\text{base}}(y_t \mid x, y_{<t}))})^{\frac{1}{|y|}}$$

$$\mathcal{L}_{\text{retain}}(\theta) = \sum_{(x,y) \in \mathcal{D}_{\text{retain}}} \prod_{t=1}^{|y|} (\frac{\min(\pi_\theta(y_t \mid x, y_{<t}), \pi_{\text{fine-tune}}(y_t \mid x, y_{<t}))}{\min(\text{sg}[\pi_\theta(y_t \mid x, y_{<t})], \pi_{\text{fine-tune}}(y_t \mid x, y_{<t}))})^{\frac{1}{|y|}}$$

where $\text{sg}[\cdot]$ is the stop-gradient operator; in other words, we treat the denominator as a constant term. Normalizing the per-token probabilities to 1 prevents gradient overflow or underflow, and clamping the retain and forget probabilities per-token helps mitigate over-unlearning.

Our total loss is then defined as:

$$\mathcal{L}(\theta) = \mathcal{L}_{\text{forget}}(\theta) - \mathcal{L}_{\text{retain}}(\theta)$$

Note that, due to the stop-gradient normalization, the scalar value of this loss is constant; we use it purely as a gradient surrogate. For logging, we compute

$$\frac{\pi_\theta(y \mid x)}{\pi_{\text{base}}(y \mid x)}$$

on the forget set to quantify the degree of unlearning.

### 3.4 PARAMETER UPDATE RULE

After computing loss over the entire training set, DAWI computes the expected change in loss from modifying each parameter.

We use a first-order approximation for $\Delta\mathcal{L}((A_\theta^k)_{i,j})$

$$\Delta\mathcal{L}((A_\theta^k)_{i,j}) \approx \frac{\partial\mathcal{L}}{\partial(A_\theta^k)_{i,j}} \Delta\beta_{i,j}^k((A_{\text{fine-tune}}^k)_{i,j} - (A_{\text{base}}^k)_{i,j})$$

During the update step, DAWI proposes to update $(A_\theta^k)_{i,j}$ by incrementing either $B_{i,j}^k$ or $C_{i,j}^k$ by 1, depending on the sign of the gradient. Thus,

$$\Delta\beta_{i,j}^k = \begin{cases} \frac{C_{i,j}^k}{C_{i,j}^k+B_{i,j}^k+1} - \frac{C_{i,j}^k}{C_{i,j}^k+B_{i,j}^k}, & \text{if } \frac{\partial\mathcal{L}}{\partial(A_\theta^k)_{i,j}}((A_{\text{fine-tune}}^k)_{i,j} - (A_{\text{base}}^k)_{i,j}) \geq 0, \\ \frac{C_{i,j}^k+1}{C_{i,j}^k+B_{i,j}^k+1} - \frac{C_{i,j}^k}{C_{i,j}^k+B_{i,j}^k}, & \text{if } \frac{\partial\mathcal{L}}{\partial(A_\theta^k)_{i,j}}((A_{\text{fine-tune}}^k)_{i,j} - (A_{\text{base}}^k)_{i,j}) < 0, \end{cases}$$

DAWI then performs the update on the accumulators, $B$ and $C$. However, since we perform a discrete update, parameters that store information unrelated to the forget set may be changed significantly, hampering the general performance of the model. Therefore, in an attempt to preserve model utility, we only update parameters that correspond to a high expected change in loss and leave other parameters unchanged. Thus, the final updates to the accumulators are:

$$\begin{cases} C_{i,j}^k \leftarrow C_{i,j}^k + 1, & \text{if } (|\Delta\mathcal{L}((A_\theta^k)_{i,j})| \geq \text{Threshold}) \text{ and } \Delta\beta_{i,j}^k \geq 0, \\ B_{i,j}^k \leftarrow B_{i,j}^k + 1, & \text{if } (|\Delta\mathcal{L}((A_\theta^k)_{i,j})| \geq \text{Threshold}) \text{ and } \Delta\beta_{i,j}^k < 0, \\ C_{i,j}^k \leftarrow C_{i,j}^k, B_{i,j}^k \leftarrow B_{i,j}^k, & \text{if } (|\Delta\mathcal{L}((A_\theta^k)_{i,j})| < \text{Threshold}) \end{cases}$$

Threshold is a hyperparameter. However, we do not tune this hyperparameter, and throughout this paper, we set it to be

$$\text{Threshold} = \tfrac{1}{5} Q_{0.99}\big( \big\{ |\Delta\mathcal{L}((A_\theta^k)_{i,j})| \big\} \big),$$

where $Q_{0.99}\{\cdot\}$ denotes the 99th percentile value of a set. Empirically, this threshold works well for all tasks.

Finally, all weights in the model are updated with the following rule:

$$(A_\theta^k)_{i,j} \leftarrow (1 - \alpha_{i,j}^k)(A_{\text{fine-tune}}^k)_{i,j} + \alpha_{i,j}^k(A_{\text{base}}^k)_{i,j}$$

## 4 UNLEARNING EXPERIMENTS

### 4.1 TOFU

In this subsection, we evaluate DAWI's performance on unlearning benchmarks. First, we demonstrate that DAWI attains strong results on the TOFU benchmark when compared to other unlearning methods. We find that with very little hyperparameter tuning, DAWI matches or exceeds the performance of a variety of strong unlearning methods.

Evaluations are primarily performed on the Forget10 split of the TOFU dataset using Llama-3.2-1B-Instruct (Grattafiori & the LLaMA Team, 2024). Forget10 contains 4000 data points with fictitious authors, with 3600 for the retain set and 400 for the forget set. We use the evaluation suite and benchmark models from Open Unlearning (Dorna et al., 2025), which is now the official repository for TOFU.

Initial testing and development of DAWI, including finding reasonable hyperparameters, was performed on the Forget1 split of TOFU, which only contains 40 data points. Therefore, DAWI's total compute budget on the Forget10 split of TOFU is comparable with the compute budget of the other benchmark methods. In fact, it is nearly an order of magnitude lower because DAWI only tunes a single hyperparameter, while other methods tune between 2 and 4 hyperparameters.

Scores are in the table below. Values are rounded to three significant figures. Best scores are in bold, and the second best are underlined. Higher is better for all scores except for PrivLeak, where closer to 0 is better. Positive PrivLeak indicates over-unlearning, and negative PrivLeak indicates under-unlearning. All metrics are from the Open Unlearning repository; our code directly calls their implementations. We benchmark IdkDPO, GradDiff, and IdkNLL using the official TOFU implementations (Maini et al., 2024). IdkDPO and IdkNLL are TOFU baselines that encourage *I don't know* responses on the forget set. Other methods include NPO (Zhang et al., 2024), SimNPO (Fan et al., 2024), RMU (Li et al., 2024), AltPO (Mekala et al., 2025), and UNDIAL (Dong et al., 2025). 400 checkpoints were evaluated in total. Additional details are in Appendix A.2.1. Results are shown in the table below.

Table 1: Scores per method on TOFU Forget10

| Method | PrivLeak | Forget Quality | Mem | Util | Priv | Overall |
|---|---|---|---|---|---|---|
| Fine-tune | -99.5 | 3.91e-22 | 0.0905 | 1.00 | 0.00650 | 0.0181 |
| Retrain | 0.0840 | 1.00 | 0.554 | 1.00 | 1.00 | 0.789 |
| | | | | | | |
| NPO | 61.8 | 4.37e-04 | 0.625 | 0.965 | 0.00790 | 0.0234 |
| IdkDPO | 57.3 | **0.994** | 0.638 | 0.977 | 0.121 | 0.276 |
| UNDIAL | -34.5 | 2.55e-08 | 0.682 | 0.872 | 0.317 | 0.520 |
| GradDiff | 60.0 | 1.02e-201 | **0.984** | 0.957 | 0.0309 | 0.0872 |
| RMU | 29.3 | 0.0446 | 0.645 | 0.924 | 0.531 | 0.664 |
| SimNPO | 36.8 | 0.813 | 0.549 | 0.989 | 0.436 | 0.585 |
| AltPO | 60.9 | 5.63e-29 | 0.678 | 0.947 | 0.0278 | 0.0779 |
| IdkNLL | -95.6 | 1.02e-13 | 0.397 | 0.859 | 0.0873 | 0.198 |
| DAWI | **4.73** | 0.581 | 0.556 | **1.00** | **0.881** | **0.763** |

## 4.2 MUD

Despite strong results on TOFU, it is unclear that DAWI retains strong model utility when $\pi_{\text{fine-tune}}$ and $\pi_{\text{base}}$ are significantly different in capability; the fine-tuned model on TOFU is only trained for a few epochs on the retain and forget sets, meaning it is not significantly more capable or generally knowledgeable than the base model. This setup does not reflect real-world deployment, where a significant portion of compute is spent during large-scale fine-tuning after pretraining. Furthermore, it is difficult to find data that accurately reflects the full set of a model's capabilities; thus, the retain set will likely omit data that exercises portions of the model's broader capabilities.

To better capture this realistic scenario, we introduce Math Unlearning Dataset 200 (MUD-200), which contains 200 synthetically generated retain and forget samples about fictitious trivia. MUD contains short trivia question-answer pairs for the retain and forget sets, and we measure model utility as accuracy on GSM8k. This is a deliberate design choice; we chose to avoid any math questions in the retain and forget sets to measure the trade-off between effective unlearning and the retention of capabilities not represented in the retain or forget sets.

To construct our fine-tuned model, we fine-tune Qwen-2.5-1.5B-Math, while using Qwen-2.5-1.5B-Instruct as the base model (Yang et al., 2024). We chose this pair of models because Qwen-2.5-1.5B-Math has undergone significantly more domain-specific training than the base instruct model, better reflecting how models are adapted in practical applications.

Due to the computational costs of training with a large set of hyperparameters, we restrict evaluation to the two strongest competitors on TOFU, RMU and SimNPO, and we add GradDiff as a baseline

comparison method. Due to the lack of a test bank of false answers, PrivLeak and Forget Quality cannot be computed. Additional details are in Appendix A.2.2.

Table 2: Scores per method on MUD-200

| Method | Mem | Util | Priv | Overall |
|---|---|---|---|---|
| Fine-tune | 0.152 | 1.00 | 0 | 0 |
| Retrain | 0.913 | 1.00 | 1.00 | 0.969 |
| DAWI | **0.907** | 0.804 | **0.435** | **0.646** |
| SimNPO | 0.842 | 0.902 | 0.0558 | 0.148 |
| GradDiff | 0.634 | 0.890 | 0.199 | 0.389 |
| RMU | 0.837 | **1.00** | 0.324 | 0.576 |

Existing unlearning benchmarks like MUSE and WMDP (Li et al., 2024) cannot be used with DAWI because their forget sets overlap with knowledge already acquired during pretraining, such as biology facts or information from Harry Potter. Since our method requires a base model that is unaware of the forget set, these benchmarks are unsuitable.

Therefore, to test DAWI's scalability, we curate MUD-20k, a larger variant of our benchmark that includes 20,000 retain and forget samples. This brings MUD to a size comparable with the largest MUSE subsets, enabling us to evaluate DAWI's ability to handle unlearning at scale.

Table 3: Scores per method on MUD-20k

| Method | Mem | Util | Priv | Overall |
|---|---|---|---|---|
| Fine-tune | 0.150 | 1.00 | 0 | 0 |
| Retrain | 0.925 | 1.00 | 1.00 | 0.974 |
| DAWI | **0.877** | 0.985 | **0.845** | **0.898** |
| SimNPO | 0.698 | 0.976 | 0.587 | 0.721 |
| GradDiff | 0.366 | **1.00** | 0.00677 | 0.0198 |
| RMU | 0.607 | **1.00** | 0.389 | 0.575 |

Interestingly, when the capability gap between the base and fine-tuned models is large, DAWI appears to perform substantially better when presented with large volumes of data. This is likely due to the discrete nature of DAWI's update rule, which makes missteps more costly compared to gradient-based methods.

### 4.3 ABLATIONS

To isolate the contributions of DAWI's optimization algorithm compared to its loss function, we train additional models on the Forget10 set.

SimNPO with clipping uses Adam and standard SimNPO loss, with the exception of clamping per-token probabilities of the forget set to the base model's probabilities to reduce over-unlearning. This is intended to test the effect of incorporating feedback from the base model on strong unlearning methods and whether it mitigates over-unlearning or improves privacy scores.

DAWI loss with Adam uses DAWI's loss function, but replaces optimization steps with Adam to test the standalone performance of DAWI's loss function.

SimNPO loss with DAWI uses SimNPO's loss function and DAWI's optimization algorithm in an attempt to gauge DAWI's performance when used with current unlearning loss functions.

DAWI without threshold sets the threshold hyperparameter to 0. This evaluates the importance of the gating mechanism.

Finally, DAWI without base runs DAWI with a randomly initialized base model to test the importance of the base model. Additional details are in Appendix A.2.3.

Table 4: Ablations on TOFU Forget10

| Method | PrivLeak | Forget Quality | Mem | Util | Priv | Overall |
|---|---|---|---|---|---|---|
| SimNPO with clipping | 20.4 | **0.641** | 0.536 | **0.990** | 0.647 | **0.676** |
| DAWI loss with Adam | -25.5 | 1.06e-239 | **0.992** | 0 | 0.991 | 0 |
| SimNPO loss with DAWI | -19.7 | 9.34e-75 | 0.879 | 1.00e-4 | **0.995** | 3.00e-4 |
| DAWI without threshold | 61.1 | 1.32e-11 | 0.758 | 0.817 | 0.0195 | 0.0557 |
| DAWI without base | **-12.9** | 1.06e-239 | 0.983 | 0 | 0.7906 | 0 |

We originally hypothesized that SimNPO loss with DAWI performs poorly due to having a smoother loss function, which may cause the threshold to include a larger number of parameters compared to DAWI loss. However, this is not the case; in fact, DAWI modifies slightly more parameters compared to SimNPO, as shown below. The number of parameters modified appears to scale logarithmically with the number of epochs.

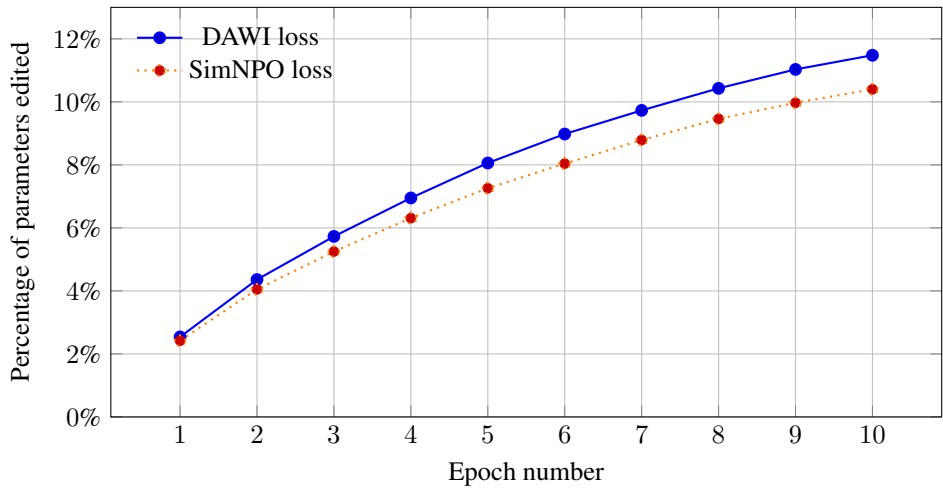

Figure 1: Total percentage of parameters modified by epoch. Both methods modify between 10% and 12% of total parameters.

## 5 MEMORY AND COMPUTE EFFICIENCY

We observe that DAWI significantly reduces GPU memory usage compared to other unlearning methods because it does not rely on a traditional optimizer. For instance, the Adam optimizer (Kingma & Ba, 2015) maintains first- and second-moment estimates for every parameter, which incurs a significant memory cost. Since updates occur once per batch, these optimizer states must remain in GPU VRAM to maintain efficient GPU utilization. In contrast, DAWI updates parameters only once per epoch, which reduces compute overhead. This design further allows reference models and accumulators to be stored in CPU memory, and since DAWI employs discrete accumulators, they can be stored in 8-bit precision without any loss in accuracy. Performance on MUD-20k is below. DAWI runs approximately 8% faster and consumes 25% less VRAM compared to other methods.

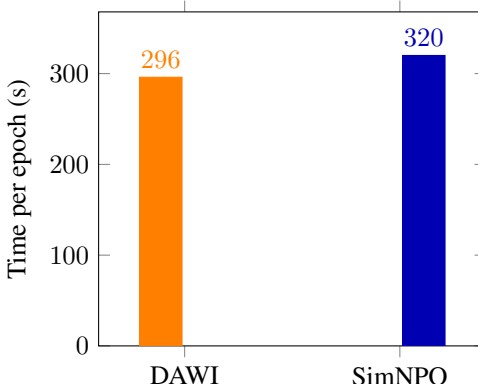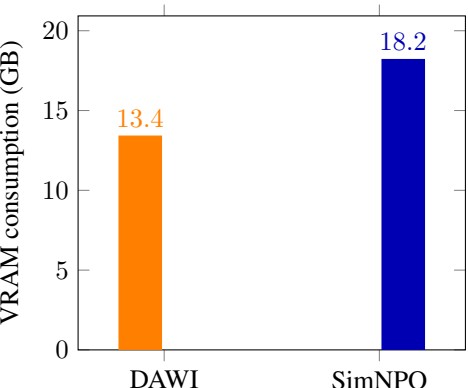

Figure 2: DAWI vs SimNPO compute and memory efficiency on MUD-20k with a batch size of 8 and models in `bfloat16`. We chose SimNPO for illustration; other methods that use Adam (GradDiff, NPO) take similar amounts of time and memory.

## 6 DISCUSSION

### 6.1 LIMITATIONS

Our ablation studies show that DAWI does not work well with loss functions designed for gradient descent and is reliant on a gating mechanism to maintain high privacy scores. Additionally, on small datasets with a large capability gap between the base and fine-tuned models, DAWI's discrete optimization steps are sensitive to noise in the dataset.

### 6.2 CONCLUSION

We introduce DAWI, a simple and computationally efficient unlearning algorithm. By constraining model parameters element-wise to be between the base and fine-tuned models and directly optimizing the probabilities of each sequence, DAWI attains significant improvements on the TOFU benchmark. DAWI demonstrates a strong capability to forget large volumes of data and retain general model utility with only 10 optimization steps. We believe that despite its flaws, DAWI is a strong unlearning method, and we hope that DAWI inspires the creation of more innovative algorithms for unlearning that go beyond modifying the loss function.

## 7 REPRODUCIBILITY STATEMENT

Training code for DAWI is anonymously released at the following GitHub mirror: `https://anonymous.4open.science/r/DAWI-CFB4/`. MUD and DAWI model checkpoints can be downloaded through anonymous HuggingFace links in the repository, and benchmark model download links are present in the repository as well. Experiments are conducted on one RTX 4090 GPU. Experiment details, including hyperparameters, for all experiments are in Appendix A.2.

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

# A APPENDIX

## A.1 ADDITIONAL ABLATION STUDIES

### A.1.1 NUMBER OF PARAMETERS EDITED PER LAYER

We also considered that SimNPO loss may cause parameters to be changed in different layers. Thus, we observe the distribution of parameters changed per layer and find that DAWI edits a more uniform proportion of parameters per layer, though both loss functions seem to peak in the middle layers.

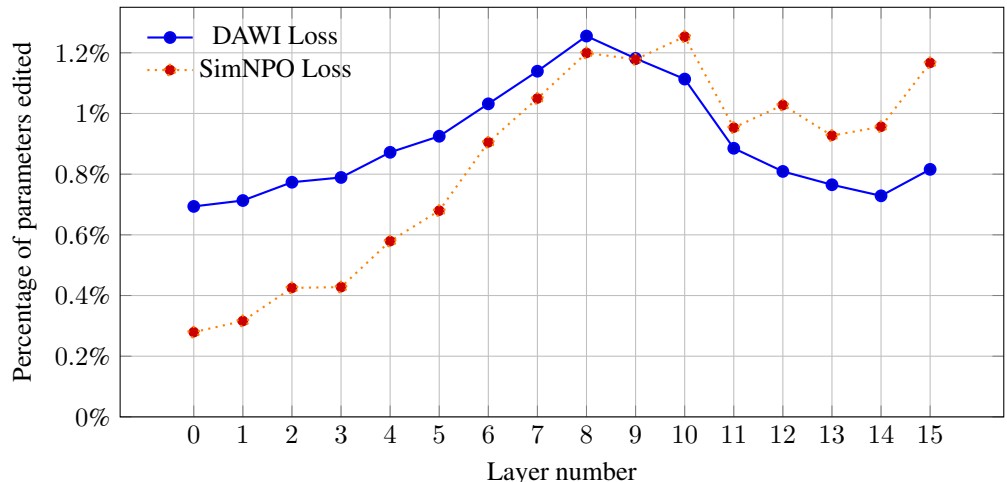

Figure 3: Layer-wise percentage of parameters edited for DAWI vs. SimNPO.

### A.1.2 OTHER ABLATIONS

Besides using DAWI's interpolation to optimize our loss function, we attempted some other commonly used optimization methods. In particular, we attempt projected gradient descent (PGD) constrained within the base and fine-tuned models' weights, with both DAWI and SimNPO loss.

Formally, if we are trying to optimize $f$ over a feasible set $\mathcal{C}$, the update rule for Projected Gradient Descent (PGD) is

$$x_{t+1} = \Pi_{\mathcal{C}}(x_t - \eta \nabla f(x_t)),$$

where $\eta$ is the learning rate and $\Pi_{\mathcal{C}}(\cdot)$ denotes the projection operator onto the set $\mathcal{C}$. In our case, we apply an element-wise clamp as our projection operator that constrains each element to be between the base and fine-tuned model's weights. Additional details in Appendix A.2.3.

Table 5: PGD scores on TOFU Forget10

| Method | PrivLeak | Forget Quality | Mem | Util | Priv | Overall |
|---|---|---|---|---|---|---|
| PGD with DAWI loss | -17.6 | 6.02e-11 | 0.501 | 0 | 0.991 | 0 |
| PGD with SimNPO loss | -96.4 | 1.02e-13 | 0.429 | 0.825 | 0.0887 | 0.204 |

## A.2 EXPERIMENT DETAILS

All results for DAWI are reported as the median of 5 runs, and we cache the probabilities of base and fine-tuned models producing a sequence on the forget and retain sets, respectively, before training; thus, we do not need to run the reference models during training.

### A.2.1 TOFU

On TOFU, we benchmark models against Open Unlearning's checkpoints and use their selection methods; the top models were selected based on the harmonic mean of their memorization and utility scores, as privacy scores are not computable without the retrained model, and the overall score was computed as the harmonic mean of all three scores. We use the Transformers library (Wolf et al., 2020) to download and run models.

All methods were trained for 10 epochs with an effective batch size of 32, and evaluated at checkpoints for epochs 5 and 10. The methods were tuned over the following hyperparameter grids, with approximately 400 checkpoints evaluated in total (using learning rate $\eta$, regularization $\alpha$, steering $\gamma$, layer index $\ell$, and $c$ is the initial value of $C_{i,j}^k$ ):

- **GradDiff & IdkNLL.** $\eta \in \{1,2,3,4,5\} \times 10^{-5}$; $\alpha \in \{1,2,5,10\}$.
- **IdkDPO, NPO, AltPO.** $\eta \in \{1,2,5\} \times 10^{-5}$; $\alpha \in \{1,2,5\}$; $\beta \in \{0.05,0.1,0.5\}$.
- **RMU.** $\eta \in \{1,2,5\} \times 10^{-5}$; $\gamma \in \{1,10,100\}$. The loss is applied to layers $\ell \in \{6,11,16\}$ of the Llama-3.2-1B model and, for each chosen $\ell$, training is restricted to layers $\{\ell-2, \ell-1, \ell\}$.
- **SimNPO.** $\eta \in \{1,2,5\} \times 10^{-5}$; $\beta \in \{3.5,4.5\}$; $\delta \in \{0,1\}$; $\theta \in \{0.125,0.25\}$.
- **UNDIAL.** $\eta \in \{1,10,30\} \times 10^{-5}$ (i.e., $1 \times 10^{-5}$, $1 \times 10^{-4}$, $3 \times 10^{-4}$); $\alpha \in \{1,2,5\}$; $\beta \in \{3,10,30\}$.
- **DAWI.** $c \in \{5,10,25\}$

The Memorization Score quantifies the degree of successful forgetting. We follow Open Unlearning's definition, where the Memorization Score is the harmonic mean (HM) of four metrics: Extraction Strength (ES) (Carlini et al., 2023), Exact memorization (EM) (Tirumala et al., 2022), Paraphrased Probability (PP) (Shi et al., 2024), and Truth Ratio (TR) (Maini et al., 2024). Each metric is inverted as $(1 - \text{metric})$, so higher scores correspond to more effective unlearning. Formally:

$$\text{Memorization Score} = \text{HM}(1 - \text{ES}, 1 - \text{EM}, 1 - \text{PP}, 1 - \text{TR}).$$

The Privacy Score evaluates resistance against membership inference attacks (MIAs). It uses four MIA metrics: LOSS, ZLib, Min-$k$, and Min-$k^{++}$. For each metric, an individual privacy score $s_{\text{MIA}}$ is computed in the range $[0,1]$, reflecting how closely the unlearned model resembles a gold-standard retrain model on that attack. Since constructing a holdout set i.i.d. from the forget set is not feasible, privacy metrics are reported after normalizing raw scores by the corresponding retrain model scores. The overall Privacy Score is the harmonic mean of the four individual scores:

$$\text{Privacy Score} = \text{HM}(s_{\text{LOSS}}, s_{\text{ZLib}}, s_{\text{Min-}k}, s_{\text{Min-}k^{++}}).$$

The Utility Score is the harmonic mean of the Gibberish Score, computed by a gibberish classifier over the model's outputs on the forget set, and Model Utility, which consists of questions about the retain set and real-world facts. Formally, the Utility Score is:

$$\text{Utility Score} = \text{HM}\big(\text{Gibberish}, \text{Model Utility}\big).$$

We report aggregate scores using the harmonic mean of normalized utility, memorization, and privacy metrics. Thus,

$$\text{Overall Score} = \text{HM}\big(s_{\text{Memorization}}, s_{\text{Privacy}}, s_{\text{Utility}}\big).$$

We compute PrivLeak as described in MUSE (Shi et al., 2024), and we include it to give an indication of whether over-unlearning has taken place. We compute forget quality, which computes the probability that samples drawn from the retrain and unlearned model are from the sample distribution, as described in TOFU (Maini et al., 2024). We include this because it was used as a benchmark for a variety of previous works, such as the NPO and SimNPO papers.

### A.2.2 MUD

Forget and memorization scores were computed in the same manner as TOFU, with one exception; Truth Ratio requires an answer bank of false choices, which are not present for MUD. Thus, we drop truth ratio from our memorization score computation. Utility scores were computed as the median accuracy on GSM8k's test split over 5 runs. Models were selected by the harmonic mean of utility and memorization scores. Due to the high computational cost of training and evaluating models, we only tune the learning rate, with $\eta \in \{1, 2, 5\} \times 10^{-5}$. All methods were trained for 10 epochs, and evaluated at epochs 5 and 10. Model utility was evaluated with VLLM (Kwon et al., 2023).

### A.2.3 ABLATION STUDIES EXPERIMENT DETAILS

For this section, we do not tune hyperparameters; instead, we use the same hyperparameters as the corresponding top-performing models on TOFU. This helps us isolate the precise changes of our ablations.

### A.3 LLM USAGE DISCLOSURE

LLMs were used for formatting and proofreading, including checking for typos, grammar mistakes, and rephrasing sentences.

