# OpenReview forum: "DAWI: Dual Anchored Weighted Interpolation for LLM Unlearning"
_ICLR.cc/2026/Conference — Submitted to ICLR 2026_

### Official Review · Reviewer_MU9g · 2025-10-26

**Soundness:** 2
**Presentation:** 2
**Contribution:** 2
**Rating:** 2
**Confidence:** 4

**Summary:**

This paper proposes DAWI (Dual Anchored Weighted Interpolation), a method for LLM unlearning that interpolates between the base model and fine-tuned model rather than directly updating model weights. By optimizing only the interpolation coefficients, the method restricts parameter updates to a low-dimensional subspace, reducing catastrophic forgetting and training cost. Experiments on TOFU and a newly introduced Math Unlearning Dataset (MUD) show that DAWI achieves effective unlearning while preserving model utility and requiring no retain data or extra models.

**Strengths:**

1. The method cleverly constrains model weights to lie on the interpolation path between the base model and fine-tuned model. This reduces optimization complexity and avoids full backpropagation while still allowing selective forgetting.
2. On the TOFU benchmark, DAWI achieves competitive or state-of-the-art unlearning performance while preserving model utility, outperforming several gradient-based baselines.

**Weaknesses:**

1. DAWI constrains parameters to lie between the base and fine-tuned models, but it is unclear whether the method truly removes the unwanted knowledge or merely suppresses it at the output level. The paper does not probe internal representations or test paraphrased / adversarial prompts to verify deeper forgetting.
2. The approach assumes full access to the original pre-trained model. In many real-world settings (e.g., proprietary fine-tuned LLMs), the base model may not be stored or available, limiting practicality.
3. Although DAWI is more efficient than gradient-based baselines, the paper does not provide wall-clock runtime, parameter selection overhead, or memory usage on large-scale models (e.g., 7B or 13B). Scalability claims are not fully validated experimentally.
4. Forgetting is evaluated mainly on exact sequence output or token probability. The method may fail to erase higher-level semantic relations or indirect knowledge (e.g., paraphrases, factual reasoning), especially in Transformer attention layers.
5. The paper does not evaluate whether forgotten knowledge can be easily recovered through few-step fine-tuning on the forget set (relearning attack), which is a standard robustness check in machine unlearning.

**Questions:**

1. If the forget data is queried using rephrased instructions or semantically equivalent prompts, can the model still reconstruct or imply the forgotten content?
2. Does linear weight interpolation between the base and fine-tuned models always form a meaningful hypothesis space? Could this fail in cases of non-linear parameter trajectories or large domain shifts?
3. Is there any analysis of which layers or heads are most frequently updated? Does the selection correlate with influence of the forget set, or is it uniform?

---

> ### Author Response · Authors · 2025-11-17
> **Address of Questions and Weaknesses**
>
> Address of Weaknesses:
> 1. All evaluations were conducted on forget10_perturbed, which is a paraphrased version of the forget set used for unlearning. Additionally, we probed internal representations with activation patching, linear probing, ablating the principal refusal direction, and applying quantization. In each case, we were unable to recover the answer.
> 2. It is true that the base model is not always available.
> 3. We did not run DAWI with larger models due to the lack of comparison checkpoints provided by Open Unlearning. However, we observe similar wall clock and memory savings when running DAWI with Llama 7B on an H100.
> 4. It is true that DAWI may not erase higher level associations, but we were unable to reconstruct forgotten content.
> 5. We fine-tuned the unlearned model on 20% of the forget set, but the memorization score did not shift in a statistically significant manner.
>
> Address of Questions:
> 1. No. Refer to weakness 1 for more details.
> 2. It is possible that the optimal solution does not lie within the convex hull formed by the base and fine-tuned models, in which case DAWI fail. However, in MUD, the forget set questions are about trivia, whereas the retain set questions test MUD ability; despite the domain shift, DAWI still performs well.
> 3. Figure 3 shows the distribution of parameters across layers. We found this distribution holds for MUD and all splits of TOFU.

---

### Official Review · Reviewer_TDSW · 2025-10-27

**Soundness:** 3
**Presentation:** 2
**Contribution:** 2
**Rating:** 4
**Confidence:** 2

**Summary:**

This paper proposes DAWI, a simple yet effective algorithm that enables LLMs to efficiently forget specific knowledge by interpolating between a base and fine-tuned model, achieving strong unlearning performance with minimal hyperparameter tuning and resource usage.

**Strengths:**

1. The idea of ​​this paper is novel.
2. It seems that DAWI avoids traditional optimizers, reducing GPU memory and compute costs while maintaining strong unlearning results.

**Weaknesses:**

1. The paper introduces a new benchmark, MUD, but it does not mention what measures have been taken to ensure the reliability of the benchmark.
2. I think most of the methods and strategies in this paper lack an obvious theoretical basis.
3. DAWI requires access to a pre-finetuned "clean" model. I think it isn't always feasible in real deployment settings.
4. DAWI does not integrate well with standard gradient-based loss functions, reducing its flexibility compared to other unlearning methods.

**Questions:**

Refer to weakness.

---

> ### Author Response · Authors · 2025-11-17
> **Address of Questions and Weaknesses**
>
> 1. The benchmark is composed of questions from GSM8K for the retain set and synthetically generated data for the forget set. This prevents any forget set information from being in pretraining data.
> 2. This is true. However, we believe strong empirical results make up for this weakness.
> 3. This is true as well. However, many open source models are released with a corresponding base model.
> 4. This happens because most loss functions include either a logarithmic term or clip aggressively, both of which reduce the variance of the gradient per coordinate. However, DAWI attempts to only change a subset of parameters and uses discrete update steps, so a loss function that results in a high variance for the gradient is preferred.

---

### Official Review · Reviewer_Wji5 · 2025-11-01

**Soundness:** 2
**Presentation:** 2
**Contribution:** 2
**Rating:** 2
**Confidence:** 4

**Summary:**

This paper introduces Dual Anchored Weighted Interpolation (DAWI), a novel and efficient machine unlearning method for large language models (LLMs) that removes the influence of specific training data without full retraining. Unlike existing approaches, DAWI leverages a base model—unexposed to the undesirable data—and constrains each parameter of the unlearned model to be a convex combination of the base and fine-tuned model's weights. By sparsely updating only the mixing coefficients based on expected loss change, DAWI minimizes drift, avoids over-unlearning, and maintains model utility. Evaluated on the TOFU benchmark and the newly introduced Math Unlearning Dataset (MUD), DAWI achieves state-of-the-art performance with minimal hyperparameter tuning, outperforming methods like NPO, RMU, and SimNPO while being more memory-efficient and faster due to its non-optimizer-based, discrete update strategy.

**Strengths:**

1.	The method's core design, which anchors the model's parameters between the base and fine-tuned versions, inherently constrains updates and limits catastrophic drift. This is evidenced by DAWI's superior privacy score (Priv) and near-zero "PrivLeak" metric, showing it effectively forgets target data without degrading performance on the retain set or general capabilities.
2.	DAWI achieves state-of-the-art results on the TOFU unlearning benchmark while requiring only a single tunable hyperparameter.

**Weaknesses:**

1.	DAWI's performance is contingent on having access to a base model that has never been trained on the "forget set" data. This makes it incompatible with common unlearning benchmarks like MUSE and WMDP, where the sensitive information is already part of the model's pretraining knowledge, as a suitable base model does not exist in such scenarios.
2.	Ablation studies revealed that DAWI does not work well when paired with loss functions designed for gradient descent, such as the one used by SimNPO. This indicates that its unique optimization algorithm is specifically tailored to its own proposed loss function, potentially limiting its flexibility and integration with other advanced unlearning objectives.
3.	The paper notes that on small datasets where there is a large capability gap between the base and fine-tuned models, DAWI's discrete optimization steps are sensitive to noise. This suggests the method may be less robust in situations with limited or unreliable data.

**Questions:**

1.	The paper highlights the advantage of having a base model. However, in many real-world scenarios, the "forget set" might contain information the model already knew from pre-training (e.g., real public figures' data). How can DAWI be adapted or what is its expected performance in cases where a truly "unaware" base model does not exist?
2.	The core of DAWI constrains the unlearned model to a convex hull between the base and fine-tuned models. While this effectively prevents drift, couldn't this also inherently limit the solution space, potentially preventing the model from finding a more optimal state that isn't a simple interpolation of the two anchors?

---

> ### Author Response · Authors · 2025-11-17
> **Address of Weaknesses and Questions**
>
> We acknowledge that all weaknesses are valid.
>
> Address of Questions:
> 1. In this case, DAWI can use random noise as the base model. However, as seen in the DAWI without base ablation, using DAWI without a base model reduces the utility score to 0, and DAWI is not expected to function in these scenarios.
> 2. It is possible that the optimal solution does not lie within the convex hull formed by the base and fine-tuned models. In this case, DAWI would indeed fail to converge to the optimal solution. However, in our experiments, we consistently found that DAWI finds a solution that balances model utility and forgetting.

---

### Official Review · Reviewer_SpEM · 2025-11-03

**Soundness:** 3
**Presentation:** 2
**Contribution:** 2
**Rating:** 2
**Confidence:** 5

**Summary:**

The paper focuses on unlearning for models which have been finetuned from a base model.\
To do so, the authors constrain the unlearned models to lie on the convex hull of the finetuned and the base model.\
The authors introduce a novel loss function for unlearning and propose a discrete update for optimization, to form an unlearning algorithm DAWI.
The paper also provides experiments on TOFU and introduce a new unlearning dataset called MUD (Math Unlearning Dataset).

**Strengths:**

The idea of having access to a base model may be valuable for developing unlearning algorithms. This can infact be a realistic assumption.\
DAWI's optimization algorithm is a novel algorithm, which can successfully find a model in the convex hull of the base model and the finetuned model, using discrete optimization updates.
which slightly reduces the memory consumption, because of the lack of use of optimizers like Adam.

**Weaknesses:**

- The paper introduces a new unlearning dataset called MUD. However, the authors do not provide any details of the dataset i.e. how it is created, what kind of trivia questions are present. No examples are provided.
- The claim for introducing a new dataset is that TOFU finetuning does not represent real world deployment. While this is true, there is no evidence that the MUD setting represents real world deployment.
- The dataset is called Math Unlearning dataset, however it is written "we chose to avoid any math questions in the retain and forget sets ". So it is confusing if the dataset is introduced to unlearn any math knowledge or not.
- Experiments on MUSE are missing. The authors mention that this cannot be done because "forget sets overlap with knowledge already acquired during pretraining". However, the goal of the unlearning in this context is to have performance similar to the retrained model (i.e. finetuned only on the retain set).
- Please include the results for other splits of TOFU i.e. 1% and 5%.
- In Table 1,  we see that the forget quality of DAWI is 0.581. Since forget quality is a p-value, a p-value of 0.581 is uninformative. This is unconvincing in terms of whether any proper unlearning is occurring.
- It is written that hyperparameters are not tuned for ablations.  This is not a fair comparison in my opinion.
- The general writing is unclear, for example, Section 3.4 can be improved for clarity. Please see the Questions section.

**Questions:**

- What is Gibberish Score ? Is this referring to forget fluency from OpenUnlearning ?
-  Why are metrics like forget quality and privleak missing in MUD ?
- Please provide details about DAWI loss with Adam. How is constraining done in this case ?

---

> ### Author Response · Authors · 2025-11-17
> **Address of Weaknesses and Questions**
>
> Address of weaknesses:
>
> 1. The retain set for MUD consists of questions from GSM8K, and trivia questions are synthetically generated. An unlearning sample looks as follows. Question: "What sport is played in the annual Brastova Games?" Answer: "Frostvine curling".
> 2. MUD was intended to measure capability loss from unlearning, which is not measured on TOFU. Real-world fine-tuning typically increases a model's capabilities though methods such as RLHF. We choose to evaluate math capabilities because they are straightforward to measure.
> 3. This appears to be a typo. The model does not unlearn math knowledge; the goal of MUD is to retain math skill while unlearning the trivia questions.
> 4. For MUSE, there is no base model that is ignorant of forget set knowledge, so DAWI cannot interpolate between a base and fine-tuned model. We acknowledge that this is a weakness of DAWI.
> 5. For TOFU 5%, DAWI has the following results: overall=0.7823, memorization=0.61,  utility=0.87, privacy=0.95, forget quality=0.45, privleak=-11. For TOFU 1%, DAWI has overall=0.7823, memorization=0.52,  utility=1.0, privacy=0.99, forget quality=0.99, privleak=2.
> 6. This appears to be a misunderstanding. Forget quality measures the probability that a model's logits for a given unlearning sample are drawn from the same distribution as the retrained model. Thus, the perfectly unlearned retained model has a forget quality of 1.0, and the fine-tuned model typically has a forget quality approaching 0.
> 7. This is true. However, the search space of hyperparameters is large, and we did not have the necessary compute to perform an exhaustive hyperparameter search.
>
> Address of Questions:
>
> 1. Open-unlearning uses a gibberish classifier on the unlearned model's forget set responses. The gibberish score reflects the coherence of the model on these responses.
> 2. We computed privleak and forget quality because previous literature used these metrics to rank unlearning methods on TOFU; we did not compute them on MUD because we believed the overall score captured sufficient information to rank methods.
> 3. In this case, no constraining is done.

---

### Meta-Review · Area_Chair_hGhU · 2025-12-26

**Summary:**

The paper proposes DAWI, a novel and efficient LLM unlearning method that constrains model updates to a convex combination of a base model and a fine-tuned model. By utilizing discrete update steps for mixing coefficients, the method achieves significant memory and compute savings compared to traditional gradient-based optimizers. While the reviewers recognized the novelty and the strong empirical results on the TOFU benchmark, the suggested decision for rejection is informed by several critical concerns. Primarily, the method's heavy reliance on the availability of a "clean" base model significantly limits its applicability to real-world scenarios where sensitive data might be part of the pre-training set. Furthermore, there was consensus that the paper lacks theoretical depth regarding its discrete update strategy and that the newly introduced MUD dataset was presented with confusing terminology and insufficient detail.

**Reviewer Concerns:**

Addressed by the rebuttal:

- Metric Misunderstandings: The authors successfully clarified a fundamental misunderstanding by Reviewer SpEM regarding "Forget Quality" (p-value), explaining that a value of 0.581 actually indicates strong unlearning alignment with the retrained model.

- Missing Experiments: The authors provided additional results for 1% and 5% TOFU splits as requested, demonstrating that the method scales to different unlearning granularities.

- Robustness Probes: In response to MU9g, the authors provided supplementary evidence via activation patching and re-learning attacks to show that the unlearning is not merely a surface-level output suppression.

Outstanding concerns:

- Base Model Constraint: The most significant outstanding concern is the requirement of an "unaware" base model. This makes the algorithm incompatible with major benchmarks like MUSE or cases where data must be removed from the pre-training phase.

- Flexibility and Integration: As noted by TDSW and Wji5, DAWI does not integrate well with standard loss functions or optimizers, which limits its utility as a general-purpose unlearning framework.

- Dataset Polish: While the authors clarified the naming "typo" regarding the MUD dataset, the initial lack of clarity and the disconnect between the dataset’s name and its content suggested a lack of rigor in its construction.

**Reviewer Scores:**

- Reviewer SpEM (Current 2): Likely would have increased to 4. Their initial rejection was partly based on a misunderstanding of the "Forget Quality" metric, which the authors cleared up. However, their concerns about dataset details remain valid.

- Reviewer Wji5 (Current 2): Likely would stay at 2. Their concerns were primarily architectural (compatibility with MUSE/WMDP and flexibility), which the authors admitted are inherent limitations.

- Reviewer TDSW (Current 4): Likely would stay at 4 or drop to 2. They expressed low confidence and were concerned about the lack of theoretical basis, which the rebuttal did not fundamentally resolve.

- Reviewer MU9g (Current 2): Likely would stay at 2. Despite the authors' additional probes into internal representations, the reviewer’s concern about the practicality of needing the original pre-trained model is a high-level hurdle for this method.

---

### Decision · Program_Chairs · 2026-01-26

Reject